# OUT-OF-DOMAIN FACT CHECKING

## ABSTRACT

Evaluating the veracity of everyday claims is time consuming and in some cases requires domain expertise. In this paper, we reveal that large commercial language models, e.g., ChatGPT or GPT4, are unable to successfully accomplish this task. We then empirically demonstrate that the commonly used fact checking pipeline, known as the retriever-reader, suffers from performance deterioration when it is trained on the labeled data from one topic (or domain) and used in another topic. Existing studies in this area mostly evaluate the transferability of fact checking systems across various platforms, e.g., Wikipedia to scientific repositories, or from one fact checking website to another one. Even in doing so, they do not step beyond pretraining models on one resource and evaluating on another resource. This calls for developing methods and techniques to make fact checking models more generalizable. Therefore, we delve into each component of the pipeline and propose algorithms to achieve this goal. We propose an adversarial algorithm to make the retriever component robust against distribution shift. Our core idea is to initially train a bi-encoder on the labeled source data, and then, to adversarially train two separate document and claim encoders using unlabeled target data. Then, we focus on the reader component and propose to train it such that it is insensitive towards the order of claims and evidence documents. Our empirical evaluations support the hypothesis that such a reader shows a higher robustness against distribution shift. To our knowledge, there is no publicly available multi-topic fact checking dataset. Thus, we propose a straightforward method to re-purpose two well-known fact checking datasets. We, then, construct eight fact checking scenarios from these datasets, and compare our model to a set of strong baseline models, including recent models that use GPT4 for generating synthetic data. Our results signify to the effectiveness of our model. Our code will be publicly available on our GitHub webpage.

## 1 INTRODUCTION

Fact checking is the task of assigning a veracity label to a given claim (Thorne & Vlachos, 2018). Automated fact checking can help with curbing misinformation in a timely manner, and can also assist professional human fact-checkers (Guo et al., 2022b). Due to the growing size of the Web, this task is increasingly becoming more challenging. The performance of fact checking systems depends on the availability of evidence resources, and this makes the large commercial language models, in many cases, unsuitable for the task. We demonstrate this by reporting an experiment in Figures 1a and 1b. We see that two existing large language models, i.e., ChatGPT and GPT 4, are unable to verify a simple claim regarding the former president of the US. At the time of carrying out this experiment, the indictment of Donald Trump was widely being discussed on news outlets, such as the Associate Press,[1] and the social media websites. Another factor that contributes to the difficulty of this task is the required technology to perform fact checking. More specifically, existing automated fact checking systems rely on a pipeline of components to retrieve evidence documents and to infer the final verdict (Guo et al., 2022b). As stated by Das et al. (2023), coordinating the components within such a pipeline is an additional burden to carry.

Given these challenges, it is desirable to know how much a trained fact checking pipeline generalizes across domains. To our knowledge, little to no work has been done to inves-

---

[1] Available at: https://apnews.com/article/trump-indicted-jan-6-investigation-special-counsel-debb59bb7a4d9f93f7e2dace01feccdc

tigate this area. There exist a few attempts (Augenstein et al., 2019; Wadden et al., 2020; Gupta & Srikumar, 2021) that report experiments on the transferability of the fact checking pipeline across various platforms, e.g., from Wikipedia to scientific repositories, or from a fact checking website to another one. While these are valuable observations, these platforms, e.g., Wikipedia and the scientific repositories, have shared domains and topics.[2] This can potentially distort certain reported conclusions, particularly, the conclusions on the generalizability of the pipeline to unseen scenarios. Additionally, the solutions proposed by existing studies to enhance generalization are very limited. To our knowledge, the only existing solution proposed by these studies is to pretrain the pipeline in one domain, and then, to use it in another domain. This further limits the toolbox of the practitioners who intend to improve their systems.

In the next section, we provide a background on the fact checking pipeline. We then report a case study to show that a pipeline trained on out-of-domain data in not as competitive as the one trained on in-domain data. We continue our study by focusing on the two primary components of the pipeline, i.e., the retriever and the reader, and propose algorithms to enhance their performance. Particularly, we use a bi-encoder dense retrieval model as the retriever, and propose an adversarial algorithm to enhance its robustness under distribution shift. We, then, exploit a previously unknown weakness of neural networks in detecting the reversal relationship between input statements, and propose an augmentation algorithm to provide the reader with more cues. The core idea of our algorithm is to train the reader such that it is insensitive towards the order of the input claims and documents.

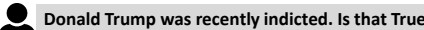

**Donald Trump was recently indicted. Is that True?**

As of my last update in September 2021, Donald Trump, the 45th President of the United States, had not been recently indicted. However, it's important to note that my information might be outdated, and I don't have access to real-time data. Legal situations and investigations can evolve over time, and new developments may have occurred since my last update.
For the most current information on Donald Trump's legal status or any potential indictments, I recommend checking reputable news sources or conducting an online search for recent news articles and updates.

(a) ChatGPT Response

**Donald Trump was recently indicted. Is that True?**

As an AI developed by OpenAI, I don't have the ability to access or retrieve real-time news updates. Please check the latest news online for the most accurate information.

(b) GPT 4 Response

Figure 1: On August 2, 2023, the Associated Press (and other news outlets) reported that Donald Trump was indicted. The questions were asked from the LLMs on August 17, 2023.

To evaluate our pipeline, we propose a straightforward method, using a public API, to re-purpose the Snopes (Hanselowski et al., 2019) and the MultiFC (Augenstein et al., 2019) fact checking datasets. We extract eight fact checking scenarios out of these two datasets, and compare our proposed components individually to the state-of-the-art techniques, including the ones that exploit GPT 4. We also demonstrate that our fact checking pipeline outperforms the pipelines that employ the best available components.

## 2 PRELIMINARY EXPERIMENTS

**Background.** Existing fact checking systems (Guo et al., 2022b; Das et al., 2023) primarily rely on two components: 1) a document retrieval model, called "retriever", and 2) a veracity prediction model, called "reader". See Figure 2a for an illustration. The retriever views the input claim as a query and returns the top evidence documents that are deemed relevant to the claim–the search is usually performed over a pre-indexed corpus. Due to the advantages that neural text retrieval models offer over the traditional token based models–such as a higher generalization and flexibility–the retriever is usually a neural document ranker (Guo et al., 2022a). As the reader, existing studies usually train a classifier over the concatenation of the retrieved documents and the given claim (Das et al., 2023).[3] As stated by Wadden et al. (2020) and Guo et al. (2022b), the veracity prediction step resembles the natural language inference task (NLI). The output of the veracity prediction

---

[2]We borrow the word "domain" in a broad sense, and the word "topic" in a narrow sense from the domain adaptation literature (Blitzer et al., 2007). In the domain adaptation literature "domain" (or "topic") refers to a particular genre (or sub-genre) of text, such as politics, business, or arts.

[3]Depending on the architecture, practitioners may add pre-processing steps, such as rationale extraction, or post-processing steps, such as justification production. We focus on the essential components.

component can be the word "Support" or the word "Refute"–depending on the system design, a third candidate output can be also added as "Neutral".

As it can be seen, developing, scaling up, and maintaining a fact checking system involves a lot of expertise, time, and budget. On the other hand, when such a system is deployed, even a small deterioration or improvement in performance can have profound impacts. Detecting an unsupported claim early enough, and then, taking timely actions on the media can be invaluable. Therefore, it is crucial to know if such a system is generalizable. In other words, if a model trained on the labeled data from one domain (i.e., source domain), demonstrates the same efficacy if it is used to verify the claims in another domain (i.e., target domain). In order to answer this question, below, we report a case study.

**Setup.** We compare the performance of in-domain fact checking compared to out-of-domain fact checking across two domains of "Miscellany" and "Politics". The claims and their evidence documents are extracted from the dataset published by Augenstein et al. (2019). The claims in this dataset do not have domain assignments. See Section 4 for our proposed method to automatically assign a domain label to each claim, and also to see a description for each domain. Each domain in this experiment has 7,900 claims, and each claim has two evidence documents. In each domain, 60% of data was used for training and 40% for testing. The claims are labeled either as Support or Refute.

In this experiment, the retriever is a bi-encoder (Karpukhin et al., 2020) pretrained using the algorithm proposed by Izacard et al. (2022). The reader is a RoBERTa-based[5] model (Liu et al., 2019) pretrained on the SNLI and MultiNLI datasets (Williams et al., 2018). Apart from these pretraining steps, all the models are fine-tuned in the source domains (using the labeled data), and then, evaluated on the target domain. We assume the target domain has no labeled data during the training. The target labels are used only for evaluation. We report Macro F1 for the classification tasks and NDCG@10 for the ranking tasks.

**Observations.** Figure 2b reports the performance of the pipeline in the in-domain scenarios compared to the out-of-domain scenarios. We see that the performance in both of the out-of-domain scenarios (i.e., Politics→Misc and Misc→Politics) is worse than their in-domain counterparts. This raises the question about the root of this performance deterioration. To reveal the cause, we report the performance of each underlying component in isolation. To evaluate the performance of the reader in isolation, we assume that the retriever perfectly returns all the relevant evidence documents. Figures 2c and 2d report the results. We see the same trend in both experiments. Both components suffer from distribution shift between the

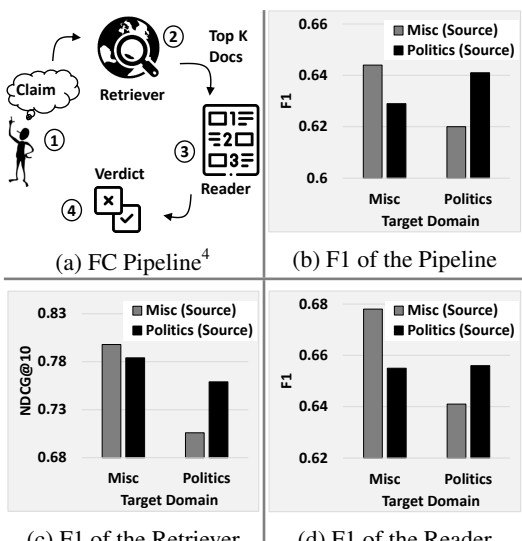

(a) FC Pipeline[4]  (b) F1 of the Pipeline

(c) F1 of the Retriever  (d) F1 of the Reader

Figure 2: **2a)** The commonly used fact checking (FC) pipeline consists of a retrieval model (called the retriever), and a veracity prediction model (called the reader). **2b)** The performance (Macro F1) of fact checking pipeline across two domains (Misc vs Politics) in two scenarios (in-domain vs out-of-domain). **2c)** The performance (NDCG@10) of the retriever across the two mentioned domains. **2d)** The performance (Macro F1) of the reader across the two domains.

in-domain and out-of-domain training. Note that the performance numbers across the scenarios and experiments are not directly comparable. For instance, one may be tempted to ask why the retriever shows a higher robustness when trained on Politics and tested on Misc, compared to the opposite scenario. Such a view may be failing to consider the properties of the claims in this dataset, such as the informativeness or the difficultness of the claims in each domain. In the next section, we for-

---

[4]The icons used in the figure have been downloaded from www.flaticon.com.

[5]We also used Deberta (He et al., 2021) in the experiments, and observed similar trends. However, we observed that this model is very unstable during the in-domain language model pretraining. Therefore, we opted for RoBERTa as the encoder to be consistent across all the experiments.

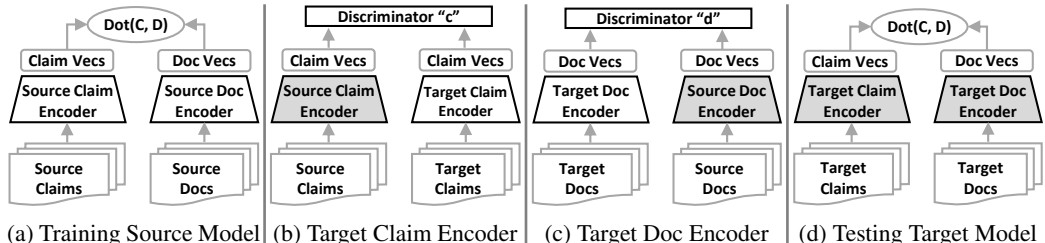

(a) Training Source Model | (b) Target Claim Encoder | (c) Target Doc Encoder | (d) Testing Target Model

Figure 3: **3a**) The source retriever is a bi-encoder, and uses dot product as the loss function. **3b**) We fix the parameters of the source claim encoder, and adversarialy train the target claim encoder to mimic the source model. This step is done using unlabeled data in the two domains. **3c**) Next, we fix the parameters of the source document encoder, and adversarialy train the target document encoder. Similarly, this step does not need labeled data. **3d**) The two trained target encoders can be used for the retrieval task in the target domain. The components that have gray outline show the inputs, outputs, and objective terms. The rest are neural networks. The parameters of the components that have gray background are fixed during training.

mally describe the problem statement, and then, we propose solutions to enhance the performance of the pipeline.

## 3 PROPOSED MODEL

### 3.1 PROBLEM STATEMENT

In the source domain $S$, we are given a set of labeled claims and their evidence documents denoted by $\{(\text{C}_i^s, y_i^s, V_i^s)\}_{i=1}^{n_s}$, where $n_s$ is the number of claims in this domain, $\text{C}_i^s$ is the *i-th* labeled claim, $y_i^s$ is the veracity of the claim–i.e., Support, Refute, or optionally Neutral–and $V_i^s$ is the set of evidence documents for supporting the assigned label. We denote the set of all the source claims by $C^s = \{\text{C}_i^s\}_{i=1}^{n_s}$, and the set of all the evidence documents by $D^s = \{\text{D}_j^s\}_{j=1}^{m_s}$, where $\text{D}_j^s$ is *j-th* evidence document, and $m_s$ is the number of evidence documents in the set. Note that $V_\bullet^s \subset D^s$. In the target domain $T$, we are given a set of unlabeled claims $C^t = \{\text{C}_i^t\}_{i=1}^{n_t}$, and a corpus of evidence documents $D^t = \{\text{D}_j^t\}_{j=1}^{m_t}$.

We opt to minimize the prediction error of the fact checking pipeline in the target domain, using the labeled data from the source domain and the unlabeled data from the target domain. Note that there is a distribution shift between the claims in the domains S and T. That is, the claims in these two domains involve different topics, discuss different entities, and refer to different events. Therefore, as we showed in the previous section, a model trained only on the out-of-domain data, usually does not perform as competitive as a model trained on in-domain data.

Following existing studies (Guo et al., 2022b; Das et al., 2023), our model adopts the pipeline illustrated in Figure 2a. We individually train each component using the labeled data from the source domain and the unlabeled data from the target domain. During testing, we plug the trained components into the pipeline to predict the veracity of the claims in the target domain. In the next section, we discuss our algorithm for training the retriever. We, then, propose our method for training the reader. We conclude the section by providing a summary of the entire training and testing procedures.

### 3.2 ADVERSARIAL TRAINING FOR EVIDENCE RETRIEVAL

We use a bi-encoder model (Karpukhin et al., 2020) in the retriever. This model consists of two encoders $f_c(\bullet)$ and $f_d(\bullet)$ to project the claims and evidence documents into low dimensional dense vectors respectively. Figure 3a illustrates the architecture of this model. To obtain a similarity score between a claim and evidence documents, a dot product operator is applied to the outputs of the encoders, i.e., for a given claim C and an evidence document D we have $sim(\text{C}, \text{D}) = f_c(\text{C})^\top \cdot f_d(\text{D})$.

To train this model in the source domain $S$, where labeled data is available, we can use the relevant evidence documents as positive examples, and the irrelevant evidence documents as negative examples. Then, we can minimize the negative log-likelihood loss term as follows:

$$\mathcal{L}_{f^s} = \sum_{i=1}^{n_s} -\log \frac{\exp(sim(\mathrm{C}_i^s, \mathrm{D}_{i+}^s))}{\exp(sim(\mathrm{C}_i^s, \mathrm{D}_{i+}^s)) + \sum_{j=1}^r \exp(sim(\mathrm{C}_i^s, \mathrm{D}_{j,i-}^s))}, \quad (1)$$

where, $n_s$ is the number of claims in the source domain, $\exp(\bullet)$ is the exponential function, $\mathrm{C}_i^s$ is the *i-th* source claim, $\mathrm{D}_{i+}^s$ is a relevant evidence document (randomly selected from the set of relevant documents $V_i^s$), and $r$ is the number of randomly selected irrelevant documents–denoted by $\mathrm{D}_{\bullet,i-}^s$. If we use stochastic gradient descent for training, we can use the irrelevant in-batch evidence documents as negative examples. The objective term is minimized with respect to the parameters of the two encoders $f_c$ and $f_d$. To test the model, we can use the similarity between a given claim and all the evidence documents, and then, can return the documents that have the highest similarity score to the claim.

Due to the lack of labeled data, the training algorithm above is not applicable in the target domain. Thus, we propose an approach to exploit unlabeled target data to train the claim and document encoders for the target domain. We begin by training a bi-encoder model in the source domain, as shown in Figure 3a. Then, we freeze the parameters of the source claim encoder, and adversarially (Goodfellow et al., 2014; Tzeng et al., 2017) train an encoder in the target domain to mimic the outputs of the source claim encoder, as shown in Figure 3b. We, then, repeat the same procedure to train a target document encoder by freezing the parameters of the source document encoder–Figure 3c. Finally, the two adversarially trained target encoders can be used to calculate the similarity between the target claims and the target evidence documents, as illustrated in Figure 3d.

The objective terms for adversarially training the target claim encoder are:

$$\mathcal{L}_{g_c} = -\mathbb{E}_{\mathrm{C}^s \sim C^s}[\log g_c(f_c^s(\mathrm{C}^s))] - \mathbb{E}_{\mathrm{C}^t \sim C^t}[\log(1 - g_c(f_c^t(\mathrm{C}^t)))], \quad (2)$$

and

$$\mathcal{L}_{f_c^t} = \mathbb{E}_{\mathrm{C}^t \sim C^t}[\log g_c(f_c^t(\mathrm{C}^t))], \quad (3)$$

where $g_c$ is the discriminator classifier for the claims, and $f_c^s$ and $f_c^t$ are the source and target claim encoders respectively. The rest of the terms were defined earlier. The two objective terms are minimized with respect to the parameters of $g_c$ and $f_c^t$ respectively. Thus, intuitively, the discriminator learns to distinguish between the claims in the source and target domains, while the target claim encoder gradually learns to produce vectors that are similar to the source vectors. Similarly, we adversarially train the target document encoder as follows:

$$\mathcal{L}_{g_d} = -\mathbb{E}_{\mathrm{D}^s \sim D^s}[\log g_d(f_d^s(\mathrm{D}^s))] - \mathbb{E}_{\mathrm{D}^t \sim D^t}[\log(1 - g_d(f_d^t(\mathrm{D}^t)))], \quad (4)$$

and

$$\mathcal{L}_{f_d^t} = \mathbb{E}_{\mathrm{D}^t \sim D^t}[\log g_d(f_d^t(\mathrm{D}^t))], \quad (5)$$

where $g_d$ is the discriminator classifier for the evidence documents, and $f_d^s$ and $f_d^t$ are the source and target document encoders respectively.

Pretraining encoders has become an integral part of dense retrieval algorithms (Karpukhin et al., 2020; Wang et al., 2022; Dai et al., 2023). Our approach for training the target claim and document encoders does not impose any restriction on the initialization of the encoders. Therefore, before training the source encoders (Figure 3a), we use the T5 model (Raffel et al., 2020) to generate a set of pseudo claims for the unlabeled evidence documents in the target domain. We, then, use this automatically generated dataset to pretrain a bi-encoder model, to be used in the training algorithm described in this section. See Section 4 for the implementation details of the pretraining step. In the next section, we discuss our algorithm for training the reader.

### 3.3 REPRESENTATION ALIGNMENT FOR VERACITY PREDICTION

To predict the veracity of a given claim, following existing studies (Wadden et al., 2020; Wright et al., 2022; Das et al., 2023), we can train a classifier on the concatenation of the corresponding evidence document and the claim–resembling the natural language inference task. If multiple evidence documents exist, we can take the average of the classifier outputs to make the final prediction. In the source domain $S$, where labeled data is available, we can employ this method. However, it is difficult to train such a classifier for the target domain because there is no labeled data in this domain. Thus, we propose an approach to train such a classifier using the labeled data in the source

domain, the unlabeled data in the target domain, and the retriever that we adversarially trained for the target domain in the previous section.

Our model is based on the distance-based discrepancy reduction training under domain shift (Ben-David et al., 2010; Long et al., 2015; Guo et al., 2020). It has been shown that (Ben-David et al., 2010) the performance of a classifier in the unlabeled target domain is a function of three terms: the performance of the classifier in the source domain, the discrepancy between the distributions of the source and target data, and the performance of the ideal classifier in both source and target domains–which depends on the properties of the data. Based on this, the common objective function for training a classifier using the labeled source data and the unlabeled target data is as follows:

$$\mathcal{L} = \frac{1}{n_s} \sum_{i=1}^{n_s} J(\theta(f_r(\mathrm{x}_i^s)), y_i^s) \ + \ \lambda \mathcal{D}(f_r(X^s), f_r(X^t)), \tag{6}$$

where $J$ is the cross entropy loss, $f_r(\bullet)$ is the data encoder, $\theta(\bullet)$ is the classifier applied to the output of the encoder, $\mathrm{x}_i^s$ is *i-th* labeled source example, and $X^s$ and $X^t$ are the sets of un-labeled source and target examples respectively. The term $\mathcal{D}$ is called the alignment loss, and reduces the discrepancy between the distributions of source and target examples after the en-coder layer. Various terms can be used as $\mathcal{D}$, such as cosine distance, Fisher linear discrimi-nant (Hastie et al., 2001), or MMD (Long et al., 2015). $\lambda > 0$ is a penalty term. We can see that the first part of the objective function performs the regular supervised training in the source domain, and the second part tries to reduce the divergence between the distributions of data in the two domains. The encoder $f_r$ appears in both parts, subjecting it to produce suit-able features for classification in the source domain (through the first part), and also, to pro-duce similar features for the data in the source and the target domains (through the second part).

In order to use Equation 6 for training our model, we need to for-mulate the vectors $\mathrm{x}_\bullet^s$, $X^s$, and $X^t$. We obtain $\mathrm{x}_\bullet^s$ and $X^s$ in the source domain by concatenating the evidence documents and their corresponding claims. Because there are no associations between the documents and the claims in the target domain, we propose to use the model trained in the previous section to retrieve the top $p$ target documents for each target claim, and then, to consider them as the evidence documents. These documents along their associated claims can be used to construct the vectors $X^t$.

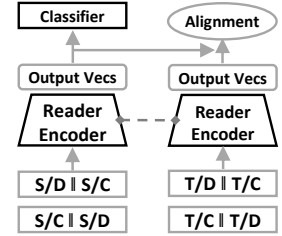

Because the trained retriever is not necessarily perfect, it may re-trieve target documents that are irrelevant to the given claim, and this can potentially propagate noise through the pipeline by pollut-ing the vectors $X^t$. In order to give the reader more cues and also provide it with more training data, we propose to augment the input data with the reverse order of itself. For instance, in the case of $\mathrm{x}_i^s$, if $\mathrm{x}_i^s = \mathrm{c}_i^s \parallel \mathrm{D}_{i+}^s$, where the symbol $\parallel$ is the concatenation operator, we then propose to also use the vector $\overline{\mathrm{x}_i^s} = \mathrm{D}_{i+}^s \parallel \mathrm{c}_i^s$ for training the reader. The augmentation can be perfromed on all the vectors in $X^s$

Figure 4: The reader model. Dashed line indicates shared parameters. S/D, S/C, T/D, and T/C stand for source doc-uments, source claims, target documents, and target claims respectively. The symbol $\parallel$ is the concatenation operator.

and $X^t$ as well. Note that in the general natural language inference task, it is not always logically true to reverse the order of the premise and the hypothesis, however, in the fact checking tack this is the case. Indeed, very recently, Berglund et al. (2023) showed that large language models are unable to infer $B{\to}A$ from $A{\to}B$.[6] This further reinforces our argument presented above.

Figure 4 shows our reader. We see that the input data is augmented with the reverse vectors. The entire model is trained using the supervised cross entropy loss and the unsupervised alignment loss.

## 3.4 TRAINING AND TESTING PROCEDURES

To train our fact checking pipeline, we use the labeled source data and the unlabeled target data in the algorithm presented in Section 3.2 to train our retriever. Then we use the trained retriever to generate pseudo-labels for the target claims. We, then, use the labeled source data along the pseudo-labeled target data to train our reader, as stated in Section 3.3.

---

[6]The paper by Berglund et al. (2023) was uploaded on arXiv on Sep 21, 2023, two days before the ICLR abstract deadline. We believe our method can be still considered novel.

While it is tempting to improve a pipeline by individually improving each component,[7] rather than proposing end-to-end solutions, this often has a downside. When our pipeline is used for testing, the improvement achieved by each component may not be fully carried over to the next step. For instance, the retriever may return better results in a particular scenario, but the reader may fail to exploit the informative evidence documents in this scenario. Another example is when the reader can potentially perform better, but the retriever fails to return informative evidence documents. In general, as stated by Domingos (2012), learning is a complex phenomenon. In order to potentially dampen the undesired effect of such cases, we add an additional step during the testing. During the testing, given an unseen target claim, the retriever is used to return the top $k$ evidence documents, and they are carried over to the reader. At this stage, instead of treating these documents as a set, we use the ranking of the documents to assign a higher weight to the top documents in making the final prediction. Therefore, instead of taking the average of the classifier to derive the prediction, we begin from the top of the list and iterate over the ranking list to generate $k$ subsets. The final prediction is made by taking the average of the predictions obtained from each subset. More formally, the final prediction is made as follows:

$$\mathcal{O} = \frac{1}{k} \sum_{i=1}^{k} \left( \frac{\sum_{j=1}^{i} \theta(f_r(C^t \parallel D_{j+}^t))}{i} \right), \tag{7}$$

where, as before, $f_r$ and $\theta$ are the reader encoder and the reader classifier, $C^t$ is the target claim at hand, and $D_{j+}^t$ is the $j$-*th* relevant document returned by the retriever. We see that the top evidence documents are present in a higher number of subsets, and therefore, have a higher weight. In the next section, we provide an overview of the experimental setup for evaluating our pipeline.

## 4 EXPERIMENTAL SETUP

We begin this section by providing an overview of the datasets used in the experiments. Then, we we briefly discuss the baselines that we compare to, and finally, we present details of the experiments. Additional information about the baselines and the training setup can be found in Appendix.

**Datasets.** We use two datasets in our experiments, the MultiFC dataset (Augenstein et al., 2019) and the Snopes dataset (Hanselowski et al., 2019). The claims in these datasets are not categorized into domains, therefore, we propose a straightforward method to automatically assign a domain name to each claim. To do so, we employ a general purpose classifier trained on a large set of categories. We opt for using the Google Content Classifier,[8] which is a multiclass model with 1,091 class labels.

| Dataset | Domain | Count | Neutral | Refute | Support |
|---------|--------|-------|---------|--------|---------|
| MultiFC | Arts | 3788 | - | 3434 | 354 |
| | Business | 1943 | - | 1007 | 936 |
| | Misc | 7968 | - | 5351 | 2617 |
| | Politics | 9350 | - | 6301 | 3049 |
| | Sensitive | 2180 | - | 1555 | 625 |
| Snopes | General | 4190 | 755 | 2643 | 792 |
| | News | 1620 | 348 | 1041 | 231 |

Table 1: The list of domains, the number of claims in each domain, and the distribution of labels in each domain for the MultiFC and Snopes datasets.

The labels assigned by the Google API are fine-grained, and in some cases, semantically close to each other. Therefore, we use a manually-crafted chart to map the Google labels to domain names. Curious readers can find this chart in Appendix A. We constructed five domains in MultiFC dataset and two domains in Snopes dataset. Table 1 reports a summary of the domains, and the distribution of the labels in each domain. The claims in the Snopes dataset are categorized into three veracity labels, whereas, the claims in the MultiFC dataset cover a much wider range of 179 labels. Due to the nature of this dataset, in many cases the labels are not easily interpretable. To make this dataset suitable for the regular fact checking task, we assign the label "Support" to those claims that are labeled as "True", and consider the rest as "Refute". We make all the claims, along their domain names, and their labels publicly available for full reproducibility.

---

[7]While this is admittedly subjective, it is tempting because usually the components in a pipeline employ methods and techniques from various study areas, and then, researchers have more opportunities to explore multiple research topics, rather than exploring a single area.

[8]Available at: https://cloud.google.com/natural-language/docs/classify-text-tutorial

To qualitatively assess the domains, and to illustrate the distribution shift between them, we report the 2D projections of the claim representations in Figures 5a and 5b. These illustrations are the outputs of a BERT encoder trained to project the claim representations, then further transformed into 2D vectors using the t-SNE technique (van der Maaten & Hinton, 2008). We observe that there is a marked shift between the distributions of each pair of the domains in both datasets. Curious readers can see sample examples of the claims and the top LDA topics of each domain in Appendix A.

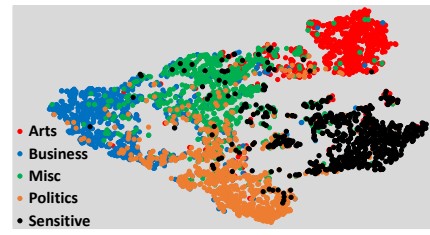

(a) MultiFC Dataset

**Baselines.** We compare our retriever with three baselines Izacard et al. (2022), Wang et al. (2022), and Dai et al. (2023). To evaluate our reader, show that it outperforms a common model proposed by existing literature, which is pretraining on the masked language model task (mlm) in the target domain, and then, finetuning in the source domain. Please see Appendix B for a discussion about the baselines and their setup.

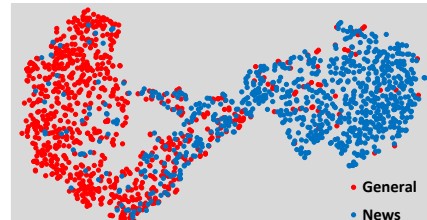

(b) Snopes Dataset

**Setup.** We follow the standard practice in domain adaptation literature (Ben-David et al., 2010) to carry out the experiments. We take several domains as source and the rest as the target domains. During training we assume we don't have access to the target labels, and use them only for testing. In the MultiFC dataset, we use the domains Misc and Politics as the source and the rest as target. We select these two as source domains because they have the highest A-distance from the rest of the domains-with 0.91 and 0.93 respectively, compared to 0.94, 0.95 and 0.96 for Business, Sensitive, and Arts respectively.[9] In Snopes , we use both domains and source and target.

Figure 5: The 2D projection of the claims in the MultiFC and Snopes datasets (using t-SNE). The vectors are the outputs of a Roberta classifier, after being trained to predict the domains. Figure best viewed in color.

| | **F1 in MultiFC** | | | | | | **F1 in Snopes** | |
| Method | M→A | M→B | M→S | P→A | P→B | P→S | G→N | N→G |
| --- | --- | --- | --- | --- | --- | --- | --- | --- |
| *cont-gpl-ft/nli-ft* | $0.580_{02}$ | $0.593_{01}$ | $0.638_{01}$ | $0.579_{02}$ | $0.595_{01}$ | $0.629_{01}$ | $0.435_{02}$ | $0.403_{02}$ |
| *cont-gpl-ft/nli-mlm-ft* | $0.581_{03}$ | $0.593_{02}$ | $0.635_{02}$ | $0.600_{02}$ | $0.590_{01}$ | $0.620_{01}$ | $0.422_{04}$ | $0.416_{01}$ |
| *cont-promp-ft/nli-ft* | $0.583_{01}$ | $0.594_{01}$ | $0.642_{01}$ | $0.586_{02}$ | $\mathbf{0.604_{00}}$ | $0.623_{01}$ | $0.434_{01}$ | $0.406_{01}$ |
| *cont-promp-ft/nli-mlm-ft* | $0.589_{03}$ | $0.594_{02}$ | $0.638_{02}$ | $0.603_{02}$ | $0.589_{02}$ | $0.619_{02}$ | $0.423_{04}$ | $0.417_{01}$ |
| *ours* | $\mathbf{0.595_{01}}$ | $\mathbf{0.605_{01}}$ | $\mathbf{0.648_{01}}$ | $\mathbf{0.615_{02}}$ | $0.603_{01}$ | $\mathbf{0.643_{01}}$ | $\mathbf{0.440_{02}}$ | $\mathbf{0.435_{01}}$ |

Table 2: Fact checking results. The sequence before "/" indicate the list of steps used in the retriever, and the sequence after "/" indicate the list of steps used in the reader. The suffix *ft* indicates finetuning on the source domain. For examples, *cont-promp-ft* means that fist Contriever is used, then Promptagator is used, and finally the model is finetuned on the source domain. For brevity, the initials of the domain names are used in the column titles.

## 5 RESULTS AND ANALYSIS

Table 2 reports the results of the fact checking pipeline across the two datasets for our model compared to the baseline methods. We observe that in all the scenarios our model is either the top performing approach, or is on a par with the best method. Tables 3 and 4 report the performance of the reader and the retriever models individually. To evaluate the reader in isolation, we assumed that the retriever return all the relevant evidence documents. Again we see that in both tasks our models outperform the baseline models. We particularly see that in both tasks, our models outperform

---

[9]A-distance (Ben-David et al., 2010) is a measure of discrepancy between two domains, and can be approximated by the error rate of a classifier trained to labels the samples from the two domains (Rai et al., 2010).

existing pretraining approaches, i.e., *nli-ft* and *cont-ft*. This confirms that our algorithms succeed in exploiting unlabeled target data.

| Method | F1 in MultiFC | | | | | | F1 in Snopes | |
|---|---|---|---|---|---|---|---|---|
| | M→A | M→B | M→S | P→A | P→B | P→S | G→N | N→G |
| *nli* | $0.443_{06}$ | $0.446_{06}$ | $0.451_{04}$ | $0.443_{06}$ | $0.446_{06}$ | $0.451_{04}$ | $0.194_{07}$ | $0.201_{06}$ |
| *nli-ft* | $0.628_{02}$ | $0.613_{00}$ | $0.646_{02}$ | $0.624_{01}$ | $0.601_{00}$ | $0.640_{01}$ | $0.454_{01}$ | $0.449_{01}$ |
| *nli-mlm-ft* | $0.614_{00}$ | $0.611_{01}$ | $0.648_{02}$ | $0.629_{03}$ | $0.600_{02}$ | $0.632_{02}$ | $0.440_{04}$ | $0.441_{00}$ |
| *ours* | $\mathbf{0.637_{02}}$ | $\mathbf{0.625_{01}}$ | $\mathbf{0.662_{01}}$ | $\mathbf{0.639_{01}}$ | $\mathbf{0.611_{02}}$ | $\mathbf{0.651_{01}}$ | $\mathbf{0.466_{01}}$ | $\mathbf{0.469_{01}}$ |

Table 3: The performance of the reader compared to the baselines. The suffix *ft* indicates finetuning on the source domain.

| Method | NDCG@10 in MultiFC | | | | | | NDCG@10 in Snopes | |
|---|---|---|---|---|---|---|---|---|
| | M→A | M→B | M→S | P→A | P→B | P→S | G→N | N→G |
| *bm25* | 0.684 | 0.723 | 0.725 | 0.684 | 0.723 | 0.725 | 0.558 | 0.638 |
| *cont-ft* | $0.673_{01}$ | $0.654_{01}$ | $0.707_{00}$ | $0.700_{00}$ | $0.663_{01}$ | $0.714_{01}$ | $0.577_{01}$ | $0.737_{00}$ |
| *cont-gpl-ft* | $0.794_{00}$ | $0.734_{01}$ | $0.784_{00}$ | $0.801_{00}$ | $0.748_{00}$ | $0.788_{00}$ | $0.642_{00}$ | $0.769_{00}$ |
| *cont-promp-ft* | $0.785_{00}$ | $0.723_{00}$ | $0.773_{01}$ | $0.796_{00}$ | $0.735_{01}$ | $0.776_{00}$ | $0.637_{00}$ | $0.766_{00}$ |
| *ours* | $\mathbf{0.803_{00}}$ | $\mathbf{0.747_{01}}$ | $\mathbf{0.793_{00}}$ | $\mathbf{0.810_{00}}$ | $\mathbf{0.757_{00}}$ | $\mathbf{0.797_{00}}$ | $\mathbf{0.647_{00}}$ | $\mathbf{0.773_{00}}$ |

Table 4: The performance of the retriever compared to the baseline models.

| Method | P→S | N→G |
|---|---|---|
| Full | 0.797 | 0.773 |
| w/o claim enc | 0.778 | 0.770 |
| w/o doc enc | 0.792 | 0.769 |

| Method | P→S | N→G |
|---|---|---|
| Full | 0.651 | 0.469 |
| w/o align | 0.638 | 0.464 |
| w/o reverse | 0.646 | 0.455 |

| Method | P→S | N→G |
|---|---|---|
| Full | 0.643 | 0.435 |
| w/o retriever | 0.640 | 0.432 |
| w/o reader | 0.632 | 0.404 |
| w/o ranking | 0.636 | 0.413 |

(a)        (b)        (c)

Table 5: Ablation studies of the proposed methods in the retriever (5a), the reader (5b), and pipeline (5c) for a use case in the MultiFC dataset (P→S) and in the Snopes dataset (N→G) .

To better understand the properties of our model, we report a series of ablation studies in both components of the pipeline, as well as the entire pipeline itself. In Table 5a, we report the performance when we omit the adversarial training of the encoders individually. We observe that each step is relatively contributing to the results. In Table 5b, we repeat the same experiment by omitting the alignment loss term and the reversal augmentation. We see that both steps are noticeably enhance the performance. Finally, to evaluate the components within the pipeline, in Table 5c, we report the performance when we disable our algorithms in the retriever, in the reader, and in the ranking the top $K$ evidence documents. We see that each component is relatively boosting the performance, however, as stated by Das et al. (2023), even though the retriever individually shows improvement, when it is within the pipeline it shows less effectiveness.

## 6 RELATED WORK

To our knowledge, our study is the first to explore automatic fact checking under domain shift. There exist a few studies that discuss the transferability of their models across fact checking websites (Augenstein et al., 2019; Wadden et al., 2020; Gupta & Srikumar, 2021). However, this can potentially distort the conclusions drawn from their results, because these platforms may have shared genre of documents and claims. Please see Appendix C for an extended discussion on related studies.

## 7 CONCLUSIONS

We studied automatic fact checking under domain shift. We showed that the large language models are unable to do the task in certain cases. Then we empirically showed that the common fact checking pipeline suffer from distribution shift, when it is trained in one domain and tested in another domain. We, then, proposed two algorithms to enhance the performance of the entire pipeline. We evaluated our model in eight scenarios and showed that in the majority of the cases our model is the top performing algorithm.

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

| Dataset | Mapped Domain | Google Content Classification Labels |
|---|---|---|
| MultiFC | Arts | /Arts & Entertainment |
| | Business | /Finance
/Business
/News/Business News |
| | Politics | /Law & Government
/News/Politics |
| | Sensitive | /Sensitive Subjects |
| | Misc | The Rest Of The Labels |
| Snopes | News | /News/Politics/Other
/News/Politics/Campaigns & Elections
/Law & Government/Government/Executive Branch
/Law & Government/Public Safety/Crime & Justice
/News/Other |
| | General | The Rest Of The Labels |

Table 6: The chart used for mapping the Google content classification labels to the domain names in each dataset.

| Dataset | Domain | Claim Example |
|---|---|---|
| MultiFC | Arts | Jennifer Lopez, Alex Rodriguez Marrying In The Spring? |
| | Business | For the first time in history the North Atlantic is empty of cargo ships in-transit |
| | Misc | Samuel Adams Set to Release New Helium Beer |
| | Politics | Hillary Clinton wore a secret earpiece during the first presidential debate of 2016 |
| | Sensitive | A man died in a meth lab explosion after attempting to light his own flatulence |
| Snopes | General | The modern image of Santa Claus was created by the Coca-Cola Company |
| | News | Donald Trump personally sent out an airplane to bring home U.S. military members stranded in Florida |

Table 7: Randomly selected claims from each domain of the MultiFc and Snopes datasets.

## A   COMPLEMENTARY REPORTS ABOUT THE DATASETS

In Table 6, we report the chart that we used to map the Google labels to the domain names in MultiFC and Snopes datasets. In Table 7, we report a set of randomly selected claims from each

| Dataset | Domain | Most Probable Topic |
|---------|--------|---------------------|
| MultiFC | Arts | fight, Matthew, Sarah, Jessica, Parker
Perry, Katy, Bloom, Orlando, Smith |
| | Business | tax, home, state, $, trust
pension, fund, work, one, say |
| | Misc | page, prayer, base, Disney, elect
turn, charge, improve, form, 2015 |
| | Politics | Meghan, Markle, Prince, Governor, political
public, day, school, record, voting |
| | Sensitive | Shooting, wear, agree, Pat
involve, media, crash, car, send |
| Snopes | General | announce, plan, California, group, Airline
document, series, Google, movie, mosque |
| | News | Donald, Trump, use, U.S., President
Clinton, Hillary, e-mail, WikiLeaks, Trump |

Table 8: Top two LDA topics for each domain of the MultiFc and Snopes datasets.

domain of the two datasets. Table 8 reports the top two topics extracted from the claims of each domain using the Latent Dirichlet Allocation algorithm (Blei et al., 2003).

## B  COMPLEMENTARY INFORMATION ABOUT THE TRAINING SETUP

**Baselines.** We evaluate our retriever from two aspects: first, we show that it is able to offer improvement over common pretraining techniques in dense retrieval for domain adaptation, and second, we show that it outperforms state-of-the-art dense retrieval methods for domain adaptation in the fact checking task. As the pretraining method, we use the model proposed by Izacard et al. (2022), called Contriever. This model is an unsupervised method based on contrastive learning by cropping spans of texts from documents and taking them as positive samples. Additionally, we compare to the models proposed by Wang et al. (2022) and Dai et al. (2023), called GPL and Promptagator. GPL uses a query generator, pretrained on the MSMarco dataset, to generate pseudo-queries for the target documents. These pseudo-queries are used to pretrain the dense retrieval model. Promptagator, is a prompt-based model that uses a large language model to generate pseudo-queries for the target documents to be used for finetuning. To have a fair comparison between the models, all of them employ an identical underlying architecture (a bi-encoder) and pre-training steps (using Contriever). The encoder in Contriever is a BERT-sized transformer-based language model, which is used in all the models. Promptagator uses a large language model for generating pseudo-labels. We use GPT 4 to generate this data. We follow the instructions stated by Dai et al. (2023) and generate 5,000 pseudo-labels for each domain, to be used for pretraining in this model. In addition to these baseline models, we also compare our model to the traditional BM25 model.

We follow the same protocol for evaluating the reader. We show that it is able to offer improvement over a relevant general domain pretraining task. For this purpose, we use the Roberta model (base variant) (Liu et al., 2019) pretrained on two NLI datasets, i.e., SNLI and MultiNLI datasets (Williams et al., 2018). Then, we also show that it outperforms a common model proposed by existing literature, which is pretraining on the masked language model task (mlm) in the target domain, and then, finetuning in the source domain. To evaluate the entire fact checking pipeline, we compare our model to the pipelines that are constructed by cross connecting two top retrievers to two top readers.

**Setup.** Our model has a few hyper-parameters. One for the coefficient of the alignment loss, and another one for the coefficient of the reverse terms–both subjects were discussed in Section 3.3. We used the domains Misc and Politics in MultiFC , and searched for the best values between $\{0.1, 0.3, 0.5, 0.7, 0.9\}$. The best values for both is 0.1. We set the value of $K$ in the reader to 10 across all the experiments–$K$ is the top documents returned by the retriever. As the alignment loss term–introduced in Equation 6, we use a metric called correlation alignment (Sun & Saenko, 2016), which measures the distance between the second-order statistics of the source and target data. For

pretraining our retriever, we use a T5 model trained on the MS-Marco dataset and generate 3 pseudo claims for each evidence document and pretrain the retriever for three epochs. We set the batch size in the retriever to 70, and in the reader to 50. We set the max sequence size for the claims to 50, and for the documents to 200. We use Adam optimizer in all the experiments. We also use gradient check-pointing for compression. We repeat all the experiments five times, and report the average results.

## C    COMPLEMENTARY RELATED WORK

There exist a few studies that investigate the transferability of the fact checking models across fact checking websites (Augenstein et al., 2019; Wadden et al., 2020; Gupta & Srikumar, 2021). Augenstein et al. (2019) compose a data set called MultiFC. This dataset was collected across multiple fact checking websites, which the authors call them "sources/domains". Their model is the standard retriever-reader pipeline, and their experiments are carried out within each website individually. Their model relies on meta-data collected from webpages. They propose no algorithm for training a model on one domain and testing on another domain. Wadden et al. (2020) compose a dataset called SciFact, collected from scientific repositories. Their model is the standard retriever-reader. To evaluate the tranferibility of their pipeline, they pretrain the pipeline on the claims extracted from wikipedia and then test it on their dataset. Thus, their solution for domain adaptation is to pretrain the pipeline on one resource and then test it on another resource; beyond this, they propose no domain adaptation method. Their study also has a shortcoming: the wikipedia claims that they use to pretrain their pipeline, may share some knowledge with the claims in their dataset. This can potentially distort their conclusions. Gupta & Srikumar (2021) compose a multilingual fact checking dataset. This dataset consists of claims, and evidence documents retrieved from Google. They use the standard pipeline, and similar to the second study, they evaluate the transferability of their pipeline by training on the data from one website and testing it on another website. Beyond this, they propose no additional solution for domain adaptation. As opposed to these studies, we delve into the two primary components of the fact checking pipeline, i.e., the retriever and the reader, and propose algorithms to enhance their robustness. Furthermore, to evaluate our model, we do not rely on comparing the results across fact checking websites, instead, we evaluate the transferability across genres of claims.

Automatic fact checking is a very active research area, interested reader can see numerous surveys published in recent years, such as the works by Oshikawa et al. (2020), Zhou & Zafarani (2021), Zeng et al. (2021), Guo et al. (2022b), Chen & Shu (2023), and Das et al. (2023). In this study, our goal is not to present an overview of existing fact checking methods, but to focus on a rather unexplored aspect of this subject, i.e., the transferability of common fact checking tools across domains. Previous studies focus on other aspects of the fact checking pipeline. For instance, Zhou et al. (2019) and Liu et al. (2020) exploit the unstructured nature of the evidence documents and propose to use graph networks for modeling the relationship between the documents. Jiang et al. (2021) concatenate all the evidence documents and use a T5 network to model the final step in the pipeline as a sequence-to-sequence problem. They report that introducing noise to the training of T5 enhances the robustness of the pipeline. Chen et al. (2022) enhance the first component of the fact checking pipeline—i.e., the retriever—by proposing a generative model to produce document titles (instead of retrieving them) to be used for retrieving evidence sentences.

There are also an overwhelming number of studies on dense text retrieval published in recent years, see the surveys by Zhao et al. (2022) and Shen et al. (2022). The method proposed by Xin et al. (2021) relies on a model called domain classifier to push the representations of source and target data points close to each other. However, as they state, because the transformation happens concurrently to the training of the retrieval encoders, it causes instability in the training. Therefore, they cache the representations of the vectors in the previous steps, and include them in their loss function. The most promising methods for domain adaptation in recent years have been those based on pseudo-query generation, such as the methods by Wang et al. (2022) and Dai et al. (2023). The first method (Wang et al., 2022) uses a pretrained model to generate pseudo-queries in the target domain. The second study (Dai et al., 2023) uses a large language model for achieving the same goal.

# D  COMPLEMENTARY EXPERIMENTS

In this section, we report two complementary experiments. First we report a comparison between our method and a fact checking pipeline that does not use any domain adaptation technique. This model is finetuned on the source domain, and then, tested on the target domain. Table 9 reports the results. We observe that in all the scenarios our model outperforms the mentioned baseline model, in some cases such as M→S and G→N by a large margin.

| Method | F1 in MultiFC | | | | | | F1 in Snopes | |
|---|---|---|---|---|---|---|---|---|
| | M→A | M→B | M→S | P→A | P→B | P→S | G→N | N→G |
| *W/O DA* | 0.590 | 0.583 | 0.607 | 0.610 | 0.573 | 0.605 | 0.383 | 0.391 |
| *ours* | **0.595** | **0.605** | **0.648** | **0.615** | **0.603** | **0.643** | **0.440** | **0.435** |

Table 9: Comparison between our model and a pipeline that does not employ domain adaptation techniques.

Second we report a comparison between our model and GPT-3. In Section 1, we empirically showed that large language models are not suitable for every day fact checking tasks, because their corpus is not regularly updated. However, it is still informative to see how these models perform on our datasets. Please note that a direct comparison between our model and a large language model is not fair, because our model requires much less hardware than these models. On the other hand, one may argue that our model has access to evidence documents. Nevertheless, given the large pretraining corpus of these models, it is also very likely that these models are pretrained directly on fact checking websites. This means that they may already have access to the ground-truth labels of the datasets in their parametric knowledge. Considering all these caveats, we report the comparison in Table 10.

| Method | F1 in MultiFC | | | | | | F1 in Snopes | |
|---|---|---|---|---|---|---|---|---|
| | M→A | M→B | M→S | P→A | P→B | P→S | G→N | N→G |
| *GPT-3* | 0.456 | 0.536 | 0.530 | 0.456 | 0.536 | 0.530 | 0.302 | 0.304 |
| *ours* | **0.595** | **0.605** | **0.648** | **0.615** | **0.603** | **0.643** | **0.440** | **0.435** |

Table 10: Comparison between our model and GPT-3. We use in-context learning to obtain the results of GPT-3. For each label in the datasets, we use two randomly selected claims along with one evidence document for each one as the in-context examples. This results in four examples in the MultiFc dataset, and six examples in the Snopes dataset. We instruct the model to return the exact labels. In the cases that the returned string is not interpretable, we assume the claim is categorized as false.

