# OpenReview forum: "Out-of-domain Fact Checking"
_ICLR.cc/2024/Conference — ICLR 2024 Conference Withdrawn Submission_

### Official Review · Reviewer_mmrP · 2023-10-20

**Soundness:** 3 good
**Presentation:** 3 good
**Contribution:** 2 fair
**Rating:** 3
**Confidence:** 4

**Summary:**

The paper discusses the challenges in evaluating the accuracy of everyday claims and highlights the limitations of large commercial language models. This work focuses more on improving the generalizability of fact-checking models by training tailored retrieval models. The authors propose an adversarial algorithm to make the retriever component more robust against distribution shifts. This method includes training a bi-encoder on labeled source data and adversarially training separate document and claim encoders using unlabeled target data.

**Strengths:**

1. The proposed problem is critical and interesting.
2. The retrieval results seem more robust and the retrieval model conducts more generalized results.

**Weaknesses:**

1. Some unsupervised methods are not compared, such as BM25. These methods have no domain shift problem.
2. The introduction describes the generalization ability of GPT3/4 on the fact verification task, but no experiments of GPT-3/4 are conducted.
3. The adversarial learning has been explored in previous work[1]. This work should discuss it.
4. Lots of related work[2-5] on fact verification is not discussed. I only list some of them.

[1] Zero-Shot Dense Retrieval with Momentum Adversarial Domain Invariant Representations.
[2] Fine-grained Fact Verification with Kernel Graph Attention Network.
[3] Exploring Listwise Evidence Reasoning with T5 for Fact Verification.
[4] GEAR: Graph-based Evidence Aggregating and Reasoning for Fact Verification.
[5] GERE: Generative Evidence Retrieval for Fact Verification

**Questions:**

Why do you not directly evaluate the fact verification performance of GPT-3/4?

---

> ### Author Response · Authors · 2023-11-22
> **Re: Official Review of Submission8415 by Reviewer mmrP**
>
> Thank you for your time. Below we provide you with a brief overview of our response. Then, you can find our main rebuttal.
>
> __Overview.__ Thank you for the comments. We improved our paper based on your feedback. We added a related work section to Appendix C. We know that reviewing papers is time consuming, so we appreciate your time. However, we found several controversial comments in your review. In the review it is said that unsupervised models have no domain shift problem, and it is also said that BM25 is not reported as a baseline in the paper. We respectfully refute these comments, because we have already reported BM25 in Table 4. Besides, as you can see there is a large performance gap between BM25 and the recent domain adaptation methods that use unlabeled target data. Our findings, as well as, the experiments reported by existing literature (Izacard et al., 2022; Wang et al., 2022; Dai et al., 2023) show that BM25 and other unsupervised models indeed suffer from domain shift. Because these models are unable to use unlabeled target data. Moreover, in the review it is said that it is not clear why we have not compared our method to the GPT models. We would like to clarify that we already answered this question in the introduction section. We reported an experiment and showed that these models are unreliable for everyday fact checking tasks and lack the needed knowledge to perform the task, because their corpus is not updated on a daily basis. In addition to this, below we report evidence that these models were trained on the ground-truth of the fact checking datasets, which makes the comparison unfair. Additionally, in the review it is suggested that we discuss the adversarial method proposed by Xin et al. (2021), and also that we add more citations to fact checking studies. We agree, however, we would like to clarify that: 1) The adversarial method proposed by Xin et al. (2021) is used as a baseline in the GPL paper (Wang et al., 2022), and is outperformed by GPL. Please note that we have used GPL as a baseline in our paper, which is a more recent method. 2) Due to space constraints, in the previous revision of our paper we discussed only the fact checking studies that explored the domain shift problem. To take into account your suggestion, we revised our paper and added a separate related work section to discuss general fact checking studies as well, please see (Appendix C). Due to the lack of space, we added this section in the appendix.

---

> > ### Comment · Reviewer_mmrP · 2023-11-22
> >
> > Indeed the authors have compared with BM25 methods. I do not think it suffers from the out-of-domain problem. It mainly due to the vocabulary mismatch problem. Thus, I can not agree with your claim.

---

> > > ### Comment · Reviewer_mmrP · 2023-11-22
> > >
> > > You said you do not compare with GPT3/4 due to the efficiency problem. But the introduction show the out-of-domain problem of LLMs. The claims are also conflict.

---

> > > > ### Author Response · Authors · 2023-11-22
> > > > **Re: Official Comment by Reviewer mmrP**
> > > >
> > > > We appreciate your reply. We respect your opinion about the GPT models.
> > > >
> > > > Yes, in the paper we stated that LLMs cannot be used for every day fact checking tasks, because their corpus is not regularly updated. In the rebuttal we are adding that even if we spend a lot of resources to regularly update them, it is unfair to compare them with our model. Because ours can be stored in a 16G memory GPU, but an LLM needs at least four 80G memory GPUs only for inference, and a lot more for training. We see no conflict in our arguments.\
> > > > We invite you and the area chair to see our discussion regarding (Comment 2) on the comparison to GPT models---in the above rebuttal letter. We believe we answered your question, and also we reported the results you asked for. But that is for the area chair to decide.

---

> > > > > ### Comment · Reviewer_mmrP · 2023-11-22
> > > > >
> > > > > I do not care the reason that you do and you do not do. The core is that you overclaim your contribution. You say LLMs face the out-of-domain problem but do not solve it.

---

> > > > > > ### Author Response · Authors · 2023-11-22
> > > > > >
> > > > > > We appreciate your reply.
> > > > > >
> > > > > > Our paper is not about LLMs, and we never made any claims about LLMs in the paper.
> > > > > >
> > > > > > We empirically demonstrated a weakness of these models to justify our effort to develop more efficient methods.

---

> > > ### Author Response · Authors · 2023-11-22
> > > **Re: Official Comment by Reviewer mmrP**
> > >
> > > We appreciate your reply. We respect what you think about BM25.
> > >
> > > Just to clarify, so we have compared to BM25, and we showed that it performs much worse than other domain adaptation baselines. As we cited above, these results are consistent with existing literature. Thus our experimental setup in this regard is complete.
> > >
> > > On a side note, the interplay between distribution shift and vocabulary mismatch, and the analysis of the performance of BM25, and the reason behind its failure are not the subject of our paper. Of course we have opinions about these subjects and may disagree with you, but they are not the subject of our study.

---

> ### Author Response · Authors · 2023-11-22
>
> __Main rebuttal.__
>
> Please find below a list of your comments. Each item is followed by a short discussion.
>
> > 1) Reviewer Comment: “Some unsupervised methods are not compared, such as BM25. These methods have no domain shift problem.”
>
> Discussion: We respectfully refute these comments. First, we would like to clarify that we reported the BM25 method in the previous revision of our paper. Please see the first row in Table 4. Second, as you can see there is a huge gap between this model and GPL and Pomptagator (second and third rows). This is due to the inability of models such as BM25 to exploit unlabeled target data. Contriever uses general datasets, which is why it performs relatively poorly. We would like to clarify that our finding regarding BM25 is consistent with the reports published by existing studies (Izacard et al., 2022; Wang et al., 2022; Dai et al., 2023).
> For further information about the baselines and experimental setup, and also how we constructed the fact checking pipelines please see (Appendix B, either revision of our paper).
>
> > 2) Reviewer Comment: “The introduction describes the generalization ability of GPT3/4 on the fact verification task, but no experiments of GPT-3/4 are conducted.”
>
> Discussion: We agree that large language models have a high generalization capability, however, in the introduction section we did not argue for their ability to perform fact checking. In (Line 6, Page 1, previous revision), we empirically showed that these models are not suitable for fact checking because their pretraining corpus is not regularly updated. Therefore, their parametric knowledge of the world is incomplete and cannot be used for everyday fact checking. Please see Figure 1, in the paper.
>
> Furthermore, it is unfair to compare a large language model, such as GPT-3, to our model. A model with the scale of GPT-3 needs about four A100 80G GPUs only for inference. Regularly finetuning this model to keep it up-to-date requires a lot more resources—not to mention what the requirements of GPT-4 are. On the other hand, our entire model can be stored in a V100 GPU with 16G of memory, and needs only a few hours for finetuning.
>
> Moreover, during our experiments we found evidence that fact checking websites are used as pretraining data for the GPT models. As the datasets used in our experiments (Hanselowski et al., 2019; Augenstein et al., 2019) are collected from fact checking websites, this makes the comparison further unfair. Because this means that these models have access to the ground-truth labels in their parametric knowledge. The evidence that we found is that, for some of the claims that we tried to make predictions for using GPT-3, the model returned pieces of texts that are specific to fact checking websites. For example, when we submitted the claim “An NPR study determined that 25 million fraudulent votes had been cast for Hillary Clinton in the 2016 presidential election.” The model returned “Pants on Fire”, which is a label used on fact checking websites to specify that a claim is completely false. This was despite the fact that we prompted the model with six in-context examples to respond with only False, True, or Neutral.
>
> Having said all these, if out of curiosity, you would still like to see the comparison, below we report the results. However, this experiment should not be taken into consideration, as the models are not comparable in terms of hardware usage, the GPT model is not suitable for everyday fact checking tasks, and there is evidence that the GPT model is trained on the ground-truth labels.
>
> ![Image](https://anonymous.4open.science/r/ICLR-2024-2C75/Screenshot%20gpt.png)
>
> If the image does not load, please go to: https://anonymous.4open.science/r/ICLR-2024-2C75/Screenshot%20gpt.png
>
> The GPT model was prompted with four in-context examples in the MultiFC dataset (for False or True), and six in-context examples in the Snopes dataset (for False, True, or Neutral). If the model output was not automatically interpretable (did not contain “true” or “neutral”), we labeled the claim as False.

---

> ### Author Response · Authors · 2023-11-22
>
> > 3) Reviewer Comment: “The adversarial learning has been explored in previous work[4]. This work should discuss it. ... Lots of related work[....] on fact verification is not discussed.”
>
> Discussion: We added a separate new section to discuss general fact checking studies—please see (Appendix C). In the previous revision of our paper we only discussed the fact checking papers that involve domain adaptation, please see (Line 1-9, Page 2, previous revision). In the newly added section, we cover a broader range of studies, including the studies that you asked us to cite. Due to the lack of space, we added this section in the appendix.
>
> Regarding the paper by Xin et al. (2021): their model was used as a baseline by Wang et al. (2022), and was shown to be weaker. We have used the most recent studies as baselines, including that of Wang et al. (2022). However, we added a discussion about the paper in the newly added related work section, please see (Appendix C). Xin et al. (2021) rely on a model called domain classifier to push the representations of source and target data points close to each other. However, because the transformation is happening concurrently to the training of the retrieval encoders, it causes instability in the training. Therefore, they cache the representations of the vectors in the previous steps, and include them in their loss function. As the results by Wang et al. (2022) show, this strategy has not been able to outperform a model like GPL. We are taking a different strategy, by initially training the encoders and then freezing them. Then training two separate encoders using two discriminators.
>
> > 1) Reviewer Question: “Why do you not directly evaluate the fact verification performance of GPT-3/4?”
>
> Discussion: We believe that we answered this question. Please see our discussion for (Comment 2).
>
> __References__
>
> [1] Izacard, Gautier, Mathilde Caron, Lucas Hosseini, Sebastian Riedel, Piotr Bojanowski, Armand Joulin, and Edouard Grave. "Unsupervised Dense Information Retrieval with Contrastive Learning." Transactions on Machine Learning Research (2022).
>
> [2] Wang, Kexin, Nandan Thakur, Nils Reimers, and Iryna Gurevych. "GPL: Generative Pseudo Labeling for Unsupervised Domain Adaptation of Dense Retrieval." In Proceedings of the 2022 Conference of the North American Chapter of the Association for Computational Linguistics: Human Language Technologies, pp. 2345-2360. 2022.
>
> [3] Dai, Zhuyun, Vincent Y. Zhao, Ji Ma, Yi Luan, Jianmo Ni, Jing Lu, Anton Bakalov, Kelvin Guu, Keith Hall, and Ming-Wei Chang. "Promptagator: Few-shot Dense Retrieval From 8 Examples." In The Eleventh International Conference on Learning Representations. 2023.
>
> [4] Xin, Ji, Chenyan Xiong, Ashwin Srinivasan, Ankita Sharma, Damien Jose, and Paul N. Bennett. "Zero-shot dense retrieval with momentum adversarial domain invariant representations." arXiv preprint arXiv:2110.07581 (2021).

---

> > ### Comment · Reviewer_mmrP · 2023-11-22
> >
> > Additional fact verification work and tasks are also not compared, such as SciFact and CLIMATE-FEVER, which also belong to specific domains. These datasets are more widely used.

---

> > > ### Author Response · Authors · 2023-11-22
> > > **Re: Official Comment by Reviewer mmrP**
> > >
> > > We appreciate your comment.
> > >
> > > We have carried out the experiments across two well-known large datasets, in eight scenarios.
> > >
> > > Yes, we considered using those two datasets as well. Note that they are small and the claims in each dataset all belong to one domain. Thus to do the experiments under domain shift, we had to carry out experiments across datasets---taking one dataset as source and another one as target. In the domain adaptation literature, this is the subject of another research topic called Label Shift [1]---the shift in the preferences of annotators. In the paper, we have made no claim about the label shift problem.
> > >
> > > On a side note, SciFact and CLIMATE-FEVER each one has about 1.5K claims. On the other hand, the smallest domain in our experiments has 1.9K claims---please see Table 1, in the paper.
> > >
> > >
> > > [1] Torralba, Antonio, and Alexei A. Efros. "Unbiased look at dataset bias." In CVPR 2011, pp. 1521-1528. IEEE, 2011.

---

### Official Review · Reviewer_FTvD · 2023-10-31

**Soundness:** 2 fair
**Presentation:** 3 good
**Contribution:** 2 fair
**Rating:** 6
**Confidence:** 4

**Summary:**

This paper discusses the challenges and potential solutions related to out-of-domain fact-checking, which involves verifying facts that are not covered by traditional domain-specific fact-checking systems. The study presents a detailed case study to demonstrate the challenges, followed by proposed methods such as adversarial training for evidence retriever and representation alignemtn for reader. The effectiveness of these methods is evaluated through comprehensive experiments and ablation study.

**Strengths:**

1. This study creatively incorporates various out-of-domain algorithms to enhance the fact-checking process.
2. The experimental setup is commendable, with a thorough ablation study that effectively showcases the efficacy of the introduced methods.
3. The paper is eloquently written, ensuring clarity and ease of comprehension for readers.

**Weaknesses:**

1. It would be beneficial to include baselines evaluating prior fact-checking algorithms (SOTA or other general ones) within the experimental setting outlined in this paper (specifically, fine-tuning on source domain and testing on target domain). This would more effectively highlight the necessity and advantages of the methods proposed for out-of-domain fact-checking.
2. Although the enhancements brought about by the proposed methods are evident in Tables 3 and 4, the cumulative improvements showcased in Table 2 appear modest.
3. Drawing a direct comparison between the proposed fact-checking system and models like ChatGPT or GPT-4 might not be entirely justifiable. Notably, the proposed system has access to evidence sources, whereas the ChatGPT or GPT-4 versions evaluated do not. It would be intriguing to see a baseline involving a GPT model (without fine-tuning) assessed on your target domain test set.

**Questions:**

1. Were experiments conducted on prior fact-checking algorithms (SOTA or other general ones), evaluated on MultiFC and Snopes using your experimental setups?
2. In the concluding lines of the 2nd paragraph in section 3.1, you state, "a model trained only on the out-of-domain data typically does not perform as competitively as a model trained on in-domain data." Could you provide clarity on how you differentiate between out-of-domain and in-domain data? Is it implied that "general" or "Misc" categories are considered out-of-domain because they amalgamate data from diverse topics? Or are the terms "out-of-domain" and "in-domain" used interchangeably throughout the paper?

---

> ### Author Response · Authors · 2023-11-22
> **Re: Official Review of Submission8415 by Reviewer FTvD**
>
> We appreciate your review. Please find below our response. We begin by providing you with an overview of the rebuttal.
>
> __Overview.__ Thank you for the comments. In the review it is suggested that we include prior SOTA models, including the model that is finetuned on the source and tested on the target. This must be a miscommunication. We would like to clarify that all of our baselines are state-of-the-art models, as presented in Izacard et al. (2022), Wang et al. (2022), and Dai et al. (2023). These models inherently include the fine-tuning step on the source data. To demonstrate how strong these baselines are, in the next section we report the results of the model that you suggested. We show that it is significantly outperformed by our baselines. Additionally, in the review it is suggested that we experiment with GPT models. While we agree that the comparison can be interesting, we would like to report that we have found evidence that the GPT models were trained on the ground-truth labels of the fact checking datasets—which makes the comparison unfair. Below we show evidence for our claim.

---

> ### Author Response · Authors · 2023-11-22
>
> __Main rebuttal.__
>
> Please find below a list of your comments, followed by a short discussion for each item.
>
> > 1) Reviewer Comment: “It would be beneficial to include baselines evaluating prior fact-checking algorithms (SOTA or other general ones) within the experimental setting outlined in this paper (specifically, fine-tuning on source domain and testing on target domain).”
>
> Discussion: below we report the performance of the model that you suggested that we include as a baseline method. This model has a bi-necoder and a reader—similar to other baseline models. The model consists of only finetuning on the labeled source data, and then testing on the target data. For brevity, we omit the full identification of the baseline names, the order of the rows matches that of Table 2 in the paper. We see that in the majority of the cases it is outperformed by all the baselines—the exception cases are M→A and P→A.
>
> ![Image](https://anonymous.4open.science/r/ICLR-2024-2C75/Screenshot%20results.png)
>
> If the image does not load, please go to: https://anonymous.4open.science/r/ICLR-2024-2C75/Screenshot%20results.png
>
> Despite the importance of the domain shift problem in fact checking, there are only three studies that are relevant to this problem, they are the studies by Augenstein et al. (2019), Wadden et al. (2020), and Gupta & Srikumar (2021)---all three were mentioned in the previous revision of our paper, please see (Line 1, Page 2, previous revision). All three studies use the standard retriever-reader pipeline. In order to evaluate the transferability of their pipelines, they propose pretraining the pipeline on the source, and then testing it on the target data—they do not propose any domain adaptation solution. However, as you can see in the table above, this approach is not as effective as the baselines that we used in the experiments.
>
> Instead of relying only on the pretraining approach to evaluate our model, we selected the state-of-the-art domain adaptation methods or the methods that are common for using unlabeled target data for each component of the pipeline. Using these methods, we constructed a set of strong fact checking pipelines that are used as baseline models, reported in Table 2 of the paper.
>
> While your doubts are understandable, which might have caused you to reduce your review scores after all the reviews were released, we justifiably claim that our study is by far the most significant work on this topic to date. We kindly request you to skim over our response to other reviewers, and see if we have resolved your doubts. Nonetheless, we are available to respond to reviewers’ questions and defend our research study. We appreciate your fair review.
>
> > 2) Reviewer Comment: “Drawing a direct comparison between the proposed fact-checking system and models like ChatGPT or GPT-4 might not be entirely justifiable. Notably, the proposed system has access to evidence sources, whereas the ChatGPT or GPT-4 versions evaluated do not. It would be intriguing to see a baseline involving a GPT model (without fine-tuning) ...”
>
> Discussion: Indeed, the opposite is true. The datasets used in our experiments (Hanselowski et al., 2019; Augenstein et al., 2019) are collected from fact checking websites. During our experiments, we found evidence that fact checking websites are used as pretraining data for the GPT models. Thus, this makes the comparison unfair, because this means that these models have access to the ground-truth labels in their parametric knowledge. The evidence that we found is that, for some of the claims that we tried to make predictions for using GPT-3, the model returned pieces of texts that are specific to fact checking websites. For example, when we submitted the claim “An NPR study determined that 25 million fraudulent votes had been cast for Hillary Clinton in the 2016 presidential election.” The model returned “Pants on Fire”, which is a label used on fact checking websites (the PolitiFact website) to specify that a claim is completely false. This was despite the fact that we prompted the model with six in-context examples to respond with only False, True, or Neutral.
>
> Additionally, and in defense of our study, in the introduction section (Figure 1, previous and current revisions), we reported an experiment showing that large language models are not suitable for everyday fact checking, because their corpus is not regularly updated. Therefore, the model is unaware of most recent events.
>
> One may argue that why not maintain a copy of a model such as GPT-3 and regularly update it. We would like to clarify that a model with the scale of GPT-3 needs about four A100 80G GPUs only for inference. Regularly finetuning this model to keep it up to date requires a lot more resources—not to mention what the requirements of GPT-4 are. On the other hand, our entire model can be stored in a V100 GPU with 16G of memory, and needs only a few hours for finetuning. Thus, the comparison is not fair.

---

> ### Author Response · Authors · 2023-11-22
>
> > 1) Reviewer Question: “Were experiments conducted on prior fact-checking algorithms (SOTA or other general ones), evaluated on MultiFC and Snopes using your experimental setups?”
>
> Discussion: We believe that we answered this question, please see our discussion for (Comment 1).
>
> > 2) Reviewer Question: “the concluding lines of the 2nd paragraph in section 3.1, you state, "a model trained only on the out-of-domain data typically does not perform as competitively as a model trained on in-domain data." Could you provide clarity on how you differentiate between out-of-domain and in-domain data? Is it implied that "general" or "Misc" categories are considered out-of-domain because they amalgamate data from diverse topics? Or are the terms "out-of-domain" and "in-domain" used interchangeably throughout the paper?”
>
> Discussion: Thank you for the question. We have not used the terms out-of-domain and in-domain interchangeably. We use these terms to characterize the relationship between a set of documents and a domain. If the distribution of the documents is the same as the distribution of the data in the domain, then the documents are called in-domain, otherwise they are called out-of-domain. The data from the two domains of “general” and “Misc” are considered out-of-domain with respect to the target domain, because this data follows a distribution different from that of the target domains. As we mentioned in the paper, please see (Footnote 2, in the paper), the data from two domains (or distributions) follows two different genres of texts.
>
> __References__
>
> [1] Izacard, Gautier, Mathilde Caron, Lucas Hosseini, Sebastian Riedel, Piotr Bojanowski, Armand Joulin, and Edouard Grave. "Unsupervised Dense Information Retrieval with Contrastive Learning." Transactions on Machine Learning Research (2022).
>
> [2] Wang, Kexin, Nandan Thakur, Nils Reimers, and Iryna Gurevych. "GPL: Generative Pseudo Labeling for Unsupervised Domain Adaptation of Dense Retrieval." In Proceedings of the 2022 Conference of the North American Chapter of the Association for Computational Linguistics: Human Language Technologies, pp. 2345-2360. 2022.
>
> [3] Dai, Zhuyun, Vincent Y. Zhao, Ji Ma, Yi Luan, Jianmo Ni, Jing Lu, Anton Bakalov, Kelvin Guu, Keith Hall, and Ming-Wei Chang. "Promptagator: Few-shot Dense Retrieval From 8 Examples." In The Eleventh International Conference on Learning Representations. 2023.

---

> ### Comment · Reviewer_FTvD · 2023-11-23
>
> Thank you for the authors' response.
>
> Regarding the discussion about baseline:
> - Thank you for providing additional experiment results of "finetuning on the labeled source data, and then testing on the target data", it would be great if the authors could add these numbers to your camera-ready version, which will help future readers of your paper to better understand the advantages of your methods and therefore strengthen your work.
> - Unfortunately, I cannot access the previous version, it says "No revisions to display." when I click the "Revision" button.
> - Although the author justifies why their paper didn't include the comparison numbers of prior arts (including Augenstein et al. (2019), Wadden et al. (2020), and Gupta & Srikumar (2021)) by stressing these methods don't propose any domain adoption technique, it would be more interesting to see if newly proposed methods outperformed prior arts on these fact-checking tasks (in same experiments setting), no matter if they proposed specific technique.
>
> Regarding the discussion about the comparison between GPT4 and the proposed method:
> - Thanks for the author's detailed explanation. I agree with the author's concern given the example. However, as the author mentions GPT models are not regularly updated, which makes it more interesting to see how they perform on unseen data. As this may involve creating a new dataset, I wouldn't expect the author to include it in this paper.
> - People are genuinely interested in how GPT models perform on these datasets, which does not necessarily reduce the contribution of this work considering the newly proposed method and GPT models were created at different levels of resources.
>
>
> Thank you for answering my questions.

---

> > ### Author Response · Authors · 2023-11-23
> >
> > Thank you for the new comments.\
> > We updated our paper. We added Appendix D, and as you suggested, we included the two additional experiments that we reported in rebuttal: 1) the comparison between our model and the finetuning model, 2) the comparison between our model and GPT-3.\
> > With the current arrangement of our paper, we are unable to add these items to the main body. For now we are adding them to appendix.
> >
> > We will also try to include the older baselines that we discussed, to provide the readers with more insight. However, this will need more time to be done.
> >
> > We are not sure what the problem is with the Openreview website that the older versions are not accessible. We are also seeing some unexpected actions, for example we are unable to directly reply to comments, and every reply is posted as a separate comment below the previous ones.
> >
> > Rebuttal period is closing tomorrow morning, we would like to once more thank you for your review and for the follow up comments. We hope that your doubts are resolved. We assure you that our work is  scientifically sound, it is innovative, and is rooted in multiple research fields, including domain adaptation literature, information retrieval literature, and automatic fact checking literature. We are also reporting significant findings about large language models.

---

### Official Review · Reviewer_5Ldi · 2023-10-31

**Soundness:** 1 poor
**Presentation:** 2 fair
**Contribution:** 1 poor
**Rating:** 3
**Confidence:** 4

**Summary:**

This paper focuses on domain adaptation for fact-checking, employing a typical unsupervised domain adaptation setting where two domains are provided: a labeled source domain and an unlabeled target domain.
The objective is to improve the performance of a source-domain trained model on the unlabeled target domain.
For the fact-checking task, the paper uses a standard retrieve-and-read pipeline comprising a bi-encoder retriever and a reader.
To adapt the bi-encoder retriever to the target domain, the paper applies adversarial training to train the bi-encoder on the unlabeled target domain, enabling it to mimic the source-domain encoder.
For adapting the reader model, the paper incorporates alignment loss in addition to cross-entropy loss during the training of the source-domain reader, integrating data from the target domain into the training process.
Moreover, the paper also proposes a  data augmentation method by switching the order of the claim and evidence document in the reader's input.
The authors also automate the conversion of two non-domain-adapted fact-checking datasets into datasets with domain labels using a fine-tuned classifier and test the proposed method on these datasets.
The results demonstrate that the proposed method outperforms the non-adapted baselines on the target domains.

**Strengths:**

1. Overall, the paper provides a clear and comprehensible description of the proposed method, particularly the setup of the entire work and the articulation of the research problem.
2. The domain adaptation method proposed in the paper is straightforward and has demonstrated better performance on the tested datasets compared to non-adapted baselines.

**Weaknesses:**

1. I believe that the innovativeness of the domain adaptation method proposed in this article is quite limited. The paper mentions that previous work on fact-checking domain adaptation tasks is scarce, but it fails to elaborate on specific limitations and lacks a detailed comparison and analysis with any domain adaptation methods. From the description in the paper, it seems that the authors have only applied standard domain adaptation techniques, such as adversarial training and alignment loss, to the common retrieve-and-read framework. Consequently, I do not see much innovation in the method, and I find the discussion of related works insufficient. The paper should at least discuss the related domain adaption works.
2. The paper’s description of data augmentation for training the reader mentions that it addresses the issues of noise due to the lack of gold documents in the target domain and provides more cues to the reader. However, I believe the data augmentation method proposed does not effectively tackle these issues. Simply swapping the order of the claim and evidence document in the input sequence does not reduce noise and might even increase it. As for providing more cues, I do not think this is achieved by merely changing the order of content in the input sequence since it does not add any additional information. The paper also lacks any empirical analysis of this specific data augmentation.
3. The quality of the created datasets is not rigorously validated. The domain labels in the dataset are labeled based on an external classifier, and the validation of the constructed domain adaptation through a simple 2D projection seems insufficient. There is even no description of how the projection is conducted in the paper.
4. The paper does not provide adequate explanations for the baselines used, such as why they are reasonable baselines. Only listing the names of the baselines is not enough. Moreover, from the description, it appears that all the baselines used are simple retrieve-and-read methods, without domain adaptation methods, making the comparison with the adapted method in the paper unreasonable. The authors should consider comparing their method with other strong adaptation methods.
5. The paper lacks an ablation analysis of the components of the proposed method. For instance, the authors do not conduct ablation studies on adversarial training or alignment loss to prove their effectiveness in domain adaptation.

**Questions:**

1. Why existing methods can only solve the fact-checking domain adaptation problem in a very limited way? What are the specific limitations?
2. Why is it believed that data augmentation can alleviate the noise issue? Why do we need to handle the order issue? Isn’t everything tested in a certain order (claim + evidence) anyway?

---

> ### Author Response · Authors · 2023-11-22
> **Re: Official Review of Submission8415 by Reviewer 5Ldi**
>
> Thank you for your review. Please see below our response. We begin by providing an overview of our rebuttal, and then, followed by the main response letter.
>
> __Overview.__ We took your comments into consideration and improved our paper. We added clarification about Figure 5 to the second paragraph of Page 8, added explanations about the baselines to Appendix B, and added a new related work section to Appendix C. We are aware that reviewing papers is time consuming, and can be challenging, so we appreciate it. However, we found several controversial comments in your review.
> - In the review it is said that we have only used non-adapted baselines. This must be a miscommunication, because all of our baselines are state-of-the-art domain adaptation methods, as presented in Izacard et al. (2022), Wang et al. (2022), and Dai et al. (2023). In the main rebuttal we will report a non-adapted fact checking model and show that in most cases, it is outperformed by all of our baselines. We agree with you, it is not reasonable to not compare with adapted baselines.
> - In the review it is said that the data augmentation does not add any new information to the data. We respectfully refute this argument. In the main rebuttal (below) we will explain a simple experiment that consists of only one data point for training, and show that our augmentation can improve the accuracy by giving additional information to the model.
> - In the review it is said that the datasets are not validated. We respectfully refute this argument, because we have adopted the protocol used in the seminal work of Hinton & Salakhutdinov (2006) for reporting data separability. We would like to clarify that we didn’t claim that we created new datasets, we have stated this in the previous revision of our paper (please Section 4, second paragraph). Instead, we have proposed a method to construct categories in already existing datasets. There are well-known methods to validate categorization and data separability, which we have adopted in the previous revision of our paper. One of these methods is data visualization, as employed by Hinton & Salakhutdinov (2006), which we used. Another method is reporting quantitative measures (Guo et al., 2020). We reported a measure called A-distance metric which has theoretical justifications (Ben-David et al., 2010) and is specifically proposed for predictions under domain shift. These were in addition to reporting a sample of claims in each domain and reporting the mappings that we used for projecting the labels. In the newer revision of our paper, we are also adding the top LDA topics for each domain as well. Please see the main rebuttal for the exact location of these items in the paper.
> - In the review it is said that the empirical analysis of the augmentation is missing. We respectfully refute this comment, please see Table 5(b), this experiment was already reported.
> - In the review it is said that ablation studies are missing, and said that no experiments on adversarial training and alignment loss are reported. We respectfully refute all these comments. Please see Tables 5(a-c), a set of ablation studies are reported, including the two experiments you mentioned.

---

> ### Author Response · Authors · 2023-11-22
>
> __Main rebuttal.__ \
> Below we provide a list of your critiques. Each item is followed by a short discussion.
>
> > 1) Reviewer Comment: “The results demonstrate that the proposed method outperforms the non-adapted baselines on the target domains. … it fails to elaborate on specific limitations and lacks a detailed comparison and analysis with any domain adaptation methods. …. the authors have only applied standard domain adaptation techniques, such as adversarial training and alignment loss, to the common retrieve-and-read framework. …. The paper should at least discuss the related domain adaptation works. … it appears that all the baselines used are simple retrieve-and-read methods, without domain adaptation methods, making the comparison with the adapted method in the paper unreasonable. The authors should consider comparing their method with other strong adaptation methods.”
>
> Discussion: We would like to clarify that all of our baseline models are state-of-the-art domain adaptation models, or they are the models that are commonly used to exploit unlabeled target data when the labeled training data is unavailable. We definitely agree with you that it is unreasonable to propose a domain adaptation model and not to compare it with other domain adaptation methods!
>
> To demonstrate the strength of our baselines, below we report a non-adapted model. Similar to other models, this model has a bi-necoder and a reader. The model consists of only finetuning on the labeled source data, and then testing on the target data—it does not use any domain adaptation technique. For brevity, we omit the full identification of the baseline names, the order of the rows matches that of Table 2 in the paper. We see that in six out of eight cases all the baselines outperform the non-adapted model—the exceptions are M→A and P→A.
>
> ![Image](https://anonymous.4open.science/r/ICLR-2024-2C75/Screenshot%20results.png)
>
> If the image does not load, please go to: https://anonymous.4open.science/r/ICLR-2024-2C75/Screenshot%20results.png
>
> - In the review it is said that it is not clear what existing domain adaptation methods are and what their limitation is: \
> Despite the importance of the domain shift problem in fact checking, there are only three studies that are relevant to this problem, they are the studies by Augenstein et al. (2019), Wadden et al. (2020), and Gupta & Srikumar (2021)---all three were mentioned in the previous revision of our paper, please see (Line 1, Page 2, previous revision). The first study (Augenstein et al., 2019) composes a data set called MultiFC. This dataset was collected across multiple fact checking websites, which the authors call them “sources/domains”. Their model is the standard retriever-reader pipeline, and their experiments are carried out within each website individually. Their model relies on meta-data collected from webpages. They propose no algorithm for training a model on one domain and testing on another domain. The second study (Wadden et al., 2020) composes a dataset called SciFact, collected from scientific repositories. Their model is the standard retriever-reader. In the experiments section of their study, they report an experiment on domain adaptation: They pretrain their pipeline on the claims extracted from wikipedia and then test it on their dataset. Thus, their solution for domain adaptation is to pretrain the pipeline on one resource and then test it on another resource; beyond this, they propose no domain adaptation method. Apart from their simple solution, their study also has a weakness: the wikipedia claims that they use to pretrain their pipeline, may share some knowledge with the claims in their dataset. This can potentially distort their conclusions. The third study (Gupta & Srikumar, 2021) composes a multilingual fact checking dataset. This dataset consists of claims, and evidence documents retrieved from Google. They use the standard pipeline, and similar to the second study, they evaluate the transferability of their pipeline by training on the data from one website and testing it on another website. Beyond this, they propose no additional solution for domain adaptation. \
> Thus, as we stated in the paper (Line 1-9, Page 2, previous revision), these studies are “limited”, as: 1) They use the claims collected from each fact checking website as the claims belonging to one domain. However, it is important to know that the claims collected from various websites may share knowledge, and do not belong to fully independent domains. 2) To tackle the domain shift problem, they propose to pretrain the pipeline on one domain and test on another domain. However, as we see in the table above, the pretraining method is not strong enough, and in most cases is outperformed by our baselines. \
> We would like to clarify that these topics were briefly discussed in the previous revision of the paper, please see (Line 1-9, Page 2, previous revision).

---

> ### Author Response · Authors · 2023-11-22
>
> - In the review it is said that related work is not discussed and said that novelty of our study is limited: \
> We added a new related work section, please see (Appendix C), and discussed a set of broad fact checking studies. Due to the lack of space, we added this section in the appendix. Based on the clarifications above, we claim that our study is by far the most innovative and comprehensive study published on this topic to date. Our contributions are: 1) We argued that evaluating the transferability of the fact checking pipeline across fact checking websites is not sufficient (see Section 1, previous revision). 2) We empirically showed that the fact checking pipeline suffers from distribution shift across domains (see Section 2, previous revision). 3) We proposed a novel adversarial domain adaptation model for the retriever component (see Section 3.1, previous revision). 4) We revealed that language models are unable to generalize from the hypothesis to the premise, if they are trained on the premise to predict the hypothesis. Based on this finding, we proposed an augmentation method to enhance the reader (see Section 3.2, previous revision). 5) Proposed a straightforward method to compose a set of domains from existing fact checking datasets and validated the quality of the extracted domains (see Section 4, previous revision). 6) Compared our model with strong domain adaptation baselines. Because the pretraining method, which is used in the previous studies and as we showed in the table above, is not strong enough, we included existing state-of-the-art domain adaptation methods for each individual component to create the baselines. This resulted in a set of fact checking pipelines, with individual components that use state-of-the-art domain adaptation techniques as the baseline models. The results show that our method outperforms these baselines in most cases (see Section 5, previous revision).
>
> > 2) Reviewer Comment: “... I believe the data augmentation method proposed does not effectively tackle these issues. Simply swapping the order of the claim and evidence document in the input sequence does not reduce noise and might even increase it. As for providing more cues, I do not think this is achieved by merely changing the order of content in the input sequence since it does not add any additional information. The paper also lacks any empirical analysis of this specific data augmentation. … Why do we need to handle the order issue? Isn’t everything tested in a certain order (claim + evidence) anyway?”
>
> Discussion: We respectfully refute these comments. First, we would like to clarify that we reported the empirical evidence for the efficacy of our augmentation algorithm in the previous revision of our paper. Please see the third row of Table 5(b). As we see there is a clear performance deterioration when we omit the augmentation from the model. Second, we would like to clarify that, as we discussed in the paper (Section 3.3), during our experiments we observed that a language model that is trained on a set of premises to evaluate their hypotheses was not able to infer the premises if it was given the hypotheses. It logically follows that if the training data is augmented with both directions, then, the model can successfully make the inference—no matter what the order of the data is, during the test phase.To support our argument, below we report an elementary example. You can actually test this example and see the result using a GPT-3 model.
>
> If the training data (in-context example) consists of the following evidence, claim, and label:
> ```
> Evidence: “MIT is the alma mater of GHI.”
> Claim: “GHI studied at MIT.”
> Label: “Is that true or false? True”
> ```
> And then the test data consists of the following evidence and claim:
> ```
> Evidence: “ABC studied at University of Illinois.”
> Claim: “University of Illinois is the alma mater of ABC.”
> Label: “Is that true or false?”
> ```
> Then, the model returns “False”, indicating that the model has not learned the relationship between “alma mater” and “study” in the training data. However, if we augment the initial training data with the following reversed example:
> ```
> Evidence: “GHI studied at MIT.”
> Claim: “MIT is the alma mater of GHI.”
> Label: “Is that true or false? True”
> ```
> Then the model correctly returns “True” for the test data point. This indicates that the augmentation gives additional cues to the model to extract the relationship between “alma mater” and “study”, and helps it to dilute the potential noise in the data.

---

> ### Author Response · Authors · 2023-11-22
>
> > 3) Reviewer Comment: “The quality of the created datasets is not rigorously validated. .... There is even no description of how the projection is conducted in the paper.”
>
> Discussion: We would like to clarify that the projection is the output of the BERT encoder classifier, transformed into a 2D space (to visualize the data points) using the t-SNE technique—we added the clarification to (the second paragraph of the section about the datasets, on Page 8, new revision). However, we respectfully refute the comment on the lack of dataset validation.
>
> We would like to clarify that we are using existing datasets, and we have not composed new datasets. This was stated in the previous revision of our paper, please see (second paragraph of Section 4). Our contribution, in this section of the paper, was to propose a method to extract a set of domains from the datasets. There are already established methods for evaluating the quality of textual domains. One of them is data visualization, which is used by the seminal work of Hinton & Salakhutdinov (2006). In Figure 5(a-b), we have used this method. Another approach is to report quantitative measures, discussed by Guo et al. (2020). In this regard, we have used a measure called A-distance, specifically designed for domain adaptation experiments, proposed by Ben-David et al. (2010)---please see (third paragraph, Page 8, previous revision). In addition to these two reports, in the previous revision of our paper we reported a sample of claims from each domain, please see (Appendix A, Table 6), and the mappings used to transfer the Google labels to domain labels, please see (Appendix A, table 7). In the current revision of our paper, we are also adding the top two LDA topics for each domain, please see (Appendix A, new revision).
>
> > 4) Reviewer Comment: “The paper does not provide adequate explanations for the baselines used, such as why they are reasonable baselines. Only listing the names of the baselines is not enough. Moreover, from the description, it appears that all the baselines used are simple retrieve-and-read methods, without domain adaptation…”
>
> Discussion: We appreciate the comment. However, we would like to clarify that, in the previous revision of our paper, we had provided more than the names of the baselines. Please see (Appendix B, previous revision), where we explained why we have selected these baselines, how they relate to our experiments, and also how we constructed the fact checking pipelines. Based on your feedback, we provided further clarifications for each baseline, please see (Appendix B, new revision).
>
> > 5) Reviewer Comment: “The paper lacks an ablation analysis of the components of the proposed method. For instance, the authors do not conduct ablation studies on adversarial training or alignment loss to prove their effectiveness in domain adaptation.”
>
> Discussion: We respectfully refute all the comments. We have reported a series of ablation studies, in the previous revision of our paper. Please see Tables 5(a-c). These also include the experiments that in the review are claimed to be missing.
>
> Please see the second and third rows of (Table 5(a), previous revision). These rows demonstrate the ablation study on the adversarial training of the claim and document encoders individually.\
> Please also see the second row of (Table 5(b), previous revision) for the results of an ablation study on the alignment loss.
>
> > 1) Reviewer Question: Two questions were asked in the review, one question about the limitations of existing fact checking methods, and another question about the need for data augmentation.
>
> Discussion: We believe that we answered both questions in the rebuttal. For the first question please see our discussion for (Comment 1), and for the second question please see our discussion for (Comment 2).
>
> __References__
>
> [1] Izacard, G, et al. "Unsupervised Dense Information Retrieval with Contrastive Learning." Tran. on Machine Learning Research (2022).
>
> [2] Wang, K, et al. "GPL: Generative Pseudo Labeling for Unsupervised Domain Adaptation of Dense Retrieval." NAACL 2022.
>
> [3] Dai, Z, et al. "Promptagator: Few-shot Dense Retrieval From 8 Examples." ICLR 2023.
>
> [4] Hinton, G, and R. Salakhutdinov. "Reducing the dimensionality of data with neural networks." science 313, no. 5786 (2006).
>
> [5] Ben-David, S, et al. "A theory of learning from different domains." Machine learning 79 (2010).
>
> [6] Guo, H, et al. "Multi-source domain adaptation for text classification via distancenet-bandits." AAAI 2020.
>
> [7] Augenstein, I, et al. "MultiFC: A Real-World Multi-Domain Dataset for Evidence-Based Fact Checking of Claims." EMNLP 2019.
>
> [8] Wadden, D, et al. "Fact or Fiction: Verifying Scientific Claims." EMNLP 2020.
>
> [9] Gupta, A, and V Srikumar. "X-Fact: A New Benchmark Dataset for Multilingual Fact Checking." ACL 2021.

---

> > ### Comment · Reviewer_5Ldi · 2023-11-23
> >
> > Thanks for the response.
> >
> > Some of my questions, such as those regarding the ablation study, have been addressed. However, I still believe that the overall contribution of this work in the domain of adaptation is quite limited.
> >
> > Regarding the baseline for dense retrieval, aside from Wang, K, et al. 2022, the other studies cited were not even designed for domain adaptation. Moreover, for the reader baseline, which is essentially a classification task, this paper fails to compare with any domain adaptation methods for classification.
> >
> > Concerning the dataset, I have yet to see a rigorous validation process, such as involving multiple annotators. Only mentioning an A distance or using visualization does not convince me of its adequacy.
> >
> > As for data augmentation, I remain skeptical that the method proposed by the authors addresses the problem they proposed. I am not convinced that simply changing the order of inputs can provide the model with more cues. And I am also worried about the generalization ability of this method.

---

### Author Response · Authors · 2023-11-22

Dear reviewers,

We appreciate your time. We are glad that you found the presentation of our work clear and comprehensible (Reviewer 5Ldi and FTvD), the research problem interesting (Reviewer mmrP), the method reasonable and straightforward (Reviewer 5Ldi), the experimental setup commendable (Reviewer FTvD), and the results generalizable and encouraging (all three reviewers).

Your feedback and comments greatly improved our work.

\- Authors